# Experiences of being treated with autologous haematopoietic stem cell transplantation for aggressive multiple sclerosis: A qualitative interview study

**Andreas Tolf**[1,2]*, **Helena Gauffin**[3,4], **Joachim Burman**[1,2], **Anne-Marie Landtblom**[1,2], **Gullvi Flensner**[5]

1 Department of Medical Sciences, *Neurology*, Uppsala University, Uppsala, Sweden, 2 Department of Neurology, Uppsala University Hospital, Uppsala, Sweden, 3 Department of Biomedical and Clinical Sciences (BKV), Faculty of Medicine and Health Sciences, Linköping University, Linköping, Sweden, 4 Department of Neurology, Linköping University Hospital, Region Östergötland, Linköping, Sweden, 5 Department of Caring Sciences, University West, Trollhättan, Sweden

* andreas.tolf@neuro.uu.se

**Data Availability Statement:** There are ethical restrictions related to participant confidentiality which prevent the public sharing of minimal data

## Abstract

### Background

Autologous haematopoietic stem cell transplantation (AHSCT) is increasingly used as a treatment for aggressive multiple sclerosis (MS) and has the potential to induce long-term remission and resolution of disease activity. Despite the extensive research on treatment outcome after AHSCT, the experience of living with MS after AHSCT has not been previously described in the scientific literature. The aim of this study was to explore long-term lived experience of people with MS treated with AHSCT.

### Methods and findings

To exclude selection bias, all persons treated with AHSCT for MS at Uppsala University Hospital, Sweden, between 2004 and 2007 (*n* = 10), were asked to participate in the study, and all accepted. Open-ended interviews were conducted, digitally recorded, transcribed verbatim, and then subjected to qualitative content analysis with an inductive approach. Five main themes emerged from the interviews: (I) being diagnosed with MS–an unpredictable existence; (II) a new treatment–a possibility for a new life; (III) AHSCT–a transition; (IV) reclaiming life; and (V) a bright future accompanied by insecurity. AHSCT was described by the participants in terms of a second chance and an opportunity for a new life. The treatment became a transition from a state of illness to a state of health, enabling a previous profound uncertainty to wane and normality to be restored. Although participants of different age and sex were included, the main limitation of this study is the relatively small number of participants. Also, the inclusion of persons from one centre alone could restrict transferability of the results.

for this study. However, the minimal data are available upon request from Head of Department, Department of Medical Sciences, Uppsala University, S-751 85 Uppsala, Sweden via phone (+46 18-611 00 00), or through their website (https://www.medsci.uu.se/About+the +Department/Contact/), for researchers who meet the criteria for access to confidential data.

**Funding:** The authors received no specific funding for this work.

**Competing interests:** The authors have declared that no competing interests exist.

**Abbreviations:** AHSCT, Autologous haematopoietic stem cell transplantation; MS, multiple sclerosis.

## Conclusions

The results give a first insight into lived experience following a highly effective induction treatment for MS, and the experience of not having MS anymore. Underpinned by previously described outcome following AHSCT, the results of this study challenge the current view on MS as a chronic disease with no possible cure.

## Introduction

Multiple sclerosis (MS) is a common auto-immune neurologic disease, affecting persons mostly in their third or fourth decade of life, but can also start in early childhood [1, 2]. Traditionally, MS is considered a life-long disease, and the prognosis can vary from a very mild disease course, to an extremely aggressive course with a rapid decline of neurological functions, leading to an untimely death [3].

The treatment era of MS started in the 1990s with the introduction of interferons [4], and since then, several new and effective treatment strategies have been incorporated in the therapeutic arsenal [5]. With the start of a treatment, the person with MS is aiming to restore control and hope, and to maintain normality [6, 7]. For the MS practitioner, it is of vital importance to recognize and gain a thorough understanding of the treatment options available, although both staying up to date with new therapies and identifying treatment failure can be challenging [8].

One of the most effective treatments for aggressive MS is high-dose immunotherapy with stem cell-support [9], often called autologous haematopoietic stem cell transplantation (AHSCT). Originally developed for haematological malignancies, it is today used also for auto-immune neurological disorders [10] and was first used in the treatment of MS in 1995 [11]. AHSCT differs from most other therapeutic strategies for MS by being performed as a one-time intervention and not as a continuous drug administration [12]. A recent randomized controlled study confirmed the effectiveness and safety [13]. Importantly, treatment-related mortality after AHSCT for MS has decreased significantly, and is today a very rare event, about 0.2% [12]. AHSCT has the potential to induce long-term remission and resolution of disease activity [14, 15], and many people with MS treated with AHSCT are hence living without active disease and without regular medication [16].

Many aspects of living with MS have been meritoriously covered in the qualitative literature, including various aspects of uncertainty, threat of a changed identity and adaption to a life with MS [17]. Despite the extensive medical research conducted to evaluate disease activity of MS after AHSCT [16], lived experience of the patients themselves in relation to the AHSCT is however, to our knowledge, not previously described. Striving for an informed and shared decision-making, knowledge about other patients' lived experience could provide valuable information for both neurologists and people with MS, for whom AHSCT could be a treatment option, when deciding upon an individualized treatment strategy [18]. Thus, the aim of this study was to explore long-term lived experience of participants undergoing AHSCT as a treatment for MS.

## Methods

An inductive approach and a qualitative methodology [19] with open-ended interviews, based on participants' lived experience, was chosen. Three of the authors (GF, HG, AML) were

previously well acquainted with this methodology. The study was conducted and the manuscript written in accordance with a COREQ checklist [20] and SRQR guidelines [21] for qualitative research.

The study was approved by the Regional Ethical Board in Uppsala (No. 2012/080/01).

All persons treated with AHSCT for MS at Uppsala University Hospital, Sweden, between 2004 and 2007 ($n = 10$), were given verbal and written information about the study (by telephone and letter, respectively); all chose to participate. The clinical outcome of this cohort and the methods for collecting this data has been previously described [14], and some of the clinical and disease-related details at time of the AHSCT and at time of the interview are presented in Table 1.

AHSCT can be conducted with various protocols of differing intensities; these are commonly classified as high, intermediate, or low intensity protocols. All ten participants underwent mobilisation of the haematopoietic stem cells with a single dose of cyclophosphamide 2 g/m$^2$ and filgrastim 5–10 µg/kg/day over 6–7 days, followed by collection of the stem cells on day 10 or 11 after the start of mobilisation. The graft was then cryopreserved without further manipulation. Nine of the participants (all except the youngest, 'Brandon') then underwent conditioning with the intermediate intensity BEAM-ATG regimen consisting of carmustine 300 mg/m$^2$, etoposide 800 mg/m$^2$, cytarabine 800 mg/m$^2$, melphalan 140 mg/m$^2$, and antithymocyte globulin 10 mg/kg. Brandon was treated with a low intensity conditioning regimen consisting of cyclophosphamide 200 mg/kg and antithymocyte globulin 6 mg/kg [14].

All interviews were conducted between 11 April 2014 and 18 February 2016 at the Department of Neurology at Uppsala University Hospital in connection with other examinations, such as lumbar puncture, magnetic resonance imaging, and positron emission tomography, as part of an extended 10-year follow-up after AHSCT for research purposes. The first participant was interviewed by JB (male, 39 years old, MD, PhD, consultant neurologist), and the other nine by AT (male, 30 years old, MD, PhD student, resident physician in neurology). None of the researchers had any previous relationship with the participants before the interviews and had not been involved in the treatment or follow-up of the participants. In one interview, the participant's partner was present in the room, although not interfering with the interview; in the other interviews, only the participant and the interviewer were present. The two authors who conducted the interviews, JB and AT, had access to information that could identify the individuals during and after the data collection, the other authors had not.

At the beginning of all interviews, the interviewers presented themselves (JB or AT) and again declared the aim of the study and the reason for the interview. Consent was reaffirmed. The participants were encouraged to tell what they remembered from the beginning of the disease course. Open-ended questions were formulated, based on the way the interviews developed, and subjects brought up by the participants, both overt and latent, were further explored with follow-up questions. No formal interview guide or questionnaire was used. In the first interview (JB), the topics presented in Table 2 emerged, topics which were covered in all interviews; when they were not brought up spontaneously by the participants, specific questions were formulated by the interviewer.

The interviews were digitally recorded, transcribed verbatim (by AT), pseudonymized and distributed to a group of interpreters (GF, HG, AT, AML). Square brackets were used to mark non-textual expressions, implicit meanings, and other information considered important for the understanding of the transcription. Although some interviews were short, the collected material provided a rich basis for analysis. Qualitative content analysis was used for the analysis, according to the method described and elaborated by Graneheim, Lundman and Lindgren [19, 22, 23]. The transcribed interviews constituted the units of analysis. After multiple readings to grasp the entirety of the content and to 'obtain a sense of the whole' [23], the individual

**Table 1. Demographic, clinical and interview data of the participants.**

| Basic data | | | Clinical data at time of AHSCT | | | | | Clinical data at time of interview | | | | Interview data | |
|---|---|---|---|---|---|---|---|---|---|---|---|---|---|
| Pseudonym | Sex | Age at MS diagnosis (y) | EDSS* | Annual relapse rate in the year preceding AHSCT | Age (y) | Length of hospitali-sation (d) | Adverse events related to acute toxicity <100 days after AHSCT | Time from AHSCT (y) | Age (y) | EDSS* | Disease activity after AHSCT | Length (min) | Inter-viewer |
| Tiffany | F | 11 | 2.0 | 3 | 14 | 23 | Febrile neutropenia. | 8 | 22 | 1.0 | No relapses. No new MRI lesions. No secondary progression. | 22 | AT |
| Christopher | M | 24 | 6.5 | 12 | 26 | 20 | Febrile neutropenia, nausea and vomiting, urinary tract infection. | 10 | 36 | 6.0 | No relapses. No new MRI lesions. No secondary progression. | 28 | AT |
| Sarah | F | 16 | 4.0 | 12 | 17 | 21 | Symptomatic typhlitis, infection with no clear cause. | 10 | 26 | 0 | No relapses. One new MRI lesion six months after AHSCT. No secondary progression. | 43 | AT |
| Nicole | F | 23 | 4.0 | 12 | 27 | 32 | Symptomatic typhlitis, sepsis, serum sickness. | 10 | 36 | 3.5 | One minor relapse after six months. One new MRI lesion after one year. Continuous immunomodulating treatment was restarted. | 21 | AT |
| Daniel | M | 24 | 7.0 | 10 | 25 | 28 | Symptomatic typhlitis. | 10 | 34 | 2.0 | Two minor relapses (after one and five years), after which continuous immunomodulating treatment was restarted. New MRI lesions after three, seven and eight years. No secondary progression. | 43 | AT |
| Lisa | F | 26 | 3.5 | 10 | 28 | 32 | Sepsis, serum sickness. | 10 | 37 | 1.5 | No relapses. No new MRI lesions. No secondary progression. | 33 | AT |
| Jennifer | F | 31 | 8.0 | 8 | 32 | 24 | Urinary tract infection, symptomatic typhlitis, oral mucositis. | 10 | 41 | 1.0 | No relapses. No new MRI lesions. No secondary progression. | 35 | JB |
| Brandon | M | 9 | 8.5 | 4 | 9 | 24 | Febrile neutropenia. | 10 | 19 | 1.0 | No relapses. No new MRI lesions. No secondary progression. | 30 | AT |
| Melissa | F | 29 | 6.5 | 4 | 33 | 29 | Sepsis, symptomatic typhlitis. | 10 | 43 | 6.0 | No relapses. No new MRI lesions. No secondary progression. | 41 | AT |
| Amy | F | 25 | 5.5 | 8 | 30 | 20 | Febrile neutropenia. | 10 | 40 | 3.0 | No relapses. No new MRI lesions. No secondary progression. | 24 | AT |

AHSCT = autologous haematopoietic stem cell transplantation, AT = Andreas Tolf, d = days, EDSS = Expanded disability status scale, F = female, JB = Joachim Burman, M = male, min = minutes, MS = multiple sclerosis, y = years.

*The Expanded disability status scale (EDSS) is a commonly used measure to quantify disability in people with multiple sclerosis. The EDSS scale ranges from 0 to 10, where 0 is indicates a normal neurological routine examination and 10 is death due to multiple sclerosis.

**Table 2. Topics covered in the interviews.**

| |
|---|
| Disease course until decision of AHSCT |
| Decision of AHSCT–information, expectations, and fears |
| AHSCT procedure |
| The role of family and friends, their situation and participation. |
| The period following AHSCT |
| View of the diagnosis of MS today |

AHSCT = autologous haematopoietic stem cell transplantation, MS = multiple sclerosis.

meaning units in the text were identified, representing a 'constellation of words or statements that relate to the same central meaning' [19]. In order to improve readability for further analysis, the meaning units were then condensed, transitioning from the verbatim transcription to a clearer, more formal expression, while preserving the original inherent meaning. Subsequently, these statements were examined and interpreted to establish their deeper meaning. This process, and interpretation of the latent content, as outlined above and exemplified in Table 3, was interpreted by three of the researchers independently (GF, HG, AT), using the software Microsoft Word (version 2016).

Interpreting the latent meaning is one of the most delicate steps in qualitative content analysis and requires an understanding of the entire context in which an individual "meaning unit" is situated. Often, such interpretation is intuitive and uncomplicated, but sometimes different researchers make different assessments and interpretations. This can involve perceiving a meaning unit differently or having differing views of the context. To strengthen trustworthiness, the analysis was therefore conducted by the three researchers entirely independently, as described above. In joint and exhaustive digital meetings, the individual analyses of the three interpreting researchers were then compared. In-depth discussions were held about important passages and interpretations that might differ, with the aim of achieving consensus on the coding, latent content and level of abstraction. AT kept a record and noted the interpretations of the latent meaning and codes that had emerged from the consensus process. At times, the audio recordings of the interviews were also played to clarify emphases and vocal expressions that could affect the interpretations. Through the joint discussions, a deeper understanding of both the individual statements, the codes, and the whole was also achieved. An example of the process with the different analysis steps is presented in Table 3.

**Table 3. Examples of the steps of the analyses from the interview with Daniel: Meaning units, condensation, interpretation of latent meaning, subtheme, and theme.**

| Meaning units | Condensation | Interpretation of latent meaning /Code | Subtheme | Theme |
|---|---|---|---|---|
| 'After a while, I saw [my doctor] once a week, just about, because I had a new symptom.' | He had to see the doctor once a week as he got new symptoms. | The progression of the disease was very rapid and dramatic. | Abnormal bodily experiences. | Being diagnosed with multiple sclerosis–an unpredictable existence. |
| 'It was anything from. . . not being able to move my legs or one of my arms, to me vomiting constantly, as soon as I was awake.' | Anything from not being able to move legs or arms to vomiting as soon as being awake. | Rapid and dramatic progression. | | |
| 'I usually don't complain; I usually accept a situation for what it is.' | Usually don't complain, usually accept a situation. | The rapid progression of the disease meant living in uncertainty. | Affecting the whole life. | |
| 'But when you don't know from one day to another if you're going to get out of bed, or if you're going to see anything, or hear anything. . .' | When you don't know from one day to another if you can get out of bed, going to see, or hear. | Uncertainty. | | |

From the deliberations in those sessions, GF and HG formulated themes and subthemes. A theme was perceived as a 'thread of meaning' [23] connecting latent content across various meaning units and codes, reflecting the text's latent content. The themes can further be subdivided into subthemes [19].

AT performed the re-contextualisation [23] by grouping the codes with their respective meaning units into the formulated themes and subthemes, using the software MAXQDA 2020 (version 20.4.2). Based on this compilation, representative citations were chosen to illustrate the themes and subthemes (AT). Citations were translated from Swedish to English by an authorized translator with experience of translating both fictional and scientific literature, and nuances in the translations were discussed in sessions with the translator. Consensus on the final themes, subthemes, and appropriate citations was reached in a meeting with all authors.

## Results

Five main themes emerged from the interviews: (I) being diagnosed with MS–an unpredictable existence, (II) a new treatment–a possibility for a new life, (III) AHSCT–a transition, (IV) reclaiming life, and (V) a bright future accompanied by insecurity.

### Theme 1: Being diagnosed with MS–an unpredictable existence

Under this theme, two sub-themes were identified: (I) abnormal bodily experiences and (II) affecting the participants' whole life and living in uncertainty.

**Subtheme 1: Abnormal bodily experiences.** The initial signs of the disease were often vividly remembered and described in great detail. Common symptoms included visual disturbances, weakness, nausea, impaired balance, and numbness. Tasks previously performed without thought or effort suddenly offered inexplicable resistance.

'I remember really well–since I play tennis–that I could not hit the ball; I kept hitting it with the edge of the racket, and I thought: "How come every stroke [fails]?"'–*Christopher*

The experience of abnormality caused great concern and resulted in a rapid involvement of health care, although symptoms that could not easily be objectively verified were often initially misunderstood, explained away, or even mistrusted by the participants themselves, by relatives, and by health care providers. The time of diagnosis was described as a life-changing, traumatic event, equal to an indisputable doom.

'I thought I was doomed. At first it was like. . . a sentence. Yes, that's it. So I had. . . it was a long time before I even articulated the diagnosis.'–*Melissa*

Due to a feeling of not being in control of the situation, as the participants could not rely on their bodily functions, a feeling of estrangement arose. By relating to one's own body as an external object, Lisa, for example, was able to ally herself with her family against her diseased body and deal with the unpleasant sensations in the legs.

'And then my feet and legs started tingling, and such things. And then, in the winter, we brought me out on the porch, and it wore off. You know, in the snow, like this. It was cold, so it wore off.'–*Tiffany*

Eventually, all participants developed an extraordinarily aggressive disease, with a high frequency of relapses, often severely disabling. Despite being given available disease-modifying

treatment at the time, new relapses evolved, and neurological deterioration proceeded. Some had remission of symptoms in-between the relapses, whereas others did not. Many had two or even more relapse symptoms going on simultaneously. In contrast to the detailed memories from the initial symptoms, this period is generally described more in summary.

'After a while, I saw [my doctor] once a week, just about, because I had a new symptom. It was anything from. . . not being able to move my legs or one of my arms, to me vomiting constantly, as soon as I was awake.'–*Daniel*

The unpredictable nature of MS, the fact that almost every imaginable symptom from the central nervous system can arise, further enhanced the participants' feeling of not being in control of the situation or of their bodies. Desperation followed their sense of being gradually stripped of their abilities, and it was difficult for them to adapt to the constantly changing conditions. A sense of alienation and confusion emerged.

'For quite a while you can think that, "oh well, people who are bound to a wheelchair, for instance, can probably cope." But once you are there, some new thing comes along. "Well, now part of my sight disappeared." And "so now my hearing is impaired", "now nobody can hear what I'm saying." You know, there was so much that I never had time to really understand what was happening, nor to accept that I got so much worse. So in the end I felt that the only thing that worked was my brain, and my hands, somewhat. [Laugher] That's how I felt.'–*Jennifer*

**Subtheme 2: Affecting the whole life and living in uncertainty.**   Life changed fundamentally, in all its aspects, as a consequence of the disease. Sudden physical and cognitive impairments reduced the participants' ability to do simple and spontaneous everyday tasks and, even more so, their ability to perform professional work. Restricted mobility obstructed and limited the freedom of movement and of action.

'So in many ways I was sort of trapped in my flat. Because the periods when I *couldn't* walk were longer than the periods when I *could* stumble along, sort of.'–*Daniel*

The participants' existence became a struggle to cope with everyday life, as they were trying to maintain normality but became dependent on the help of others, for example relatives, heath care, and public services. Often, however, it was the immediate family who first had to take the primary responsibility for the difficulties that arose, and to some extent even for nursing. This situation put integrity to the test, especially for young people, having just entered adulthood. Siblings of the children with MS could have a feeling of being overlooked and forced to make sacrifices, as a result of the family's existence being largely dominated by healthcare contacts and care; a topic that can be difficult to raise, even as many years had pass. In the adults with MS, the inability to cope with life without practical and psychological support from their families gave rise to profound feelings of guilt over being a burden, sometimes with dramatic expressions.

'God, what my family have had to endure! For a while I had just about decided: "I think I'll die now. I can't take it anymore." So I called them at night, saying: "I'll die tonight." I thought that if I only decided to stop breathing, I'd die. But after five minutes, somebody was there, sitting by my side. But to my mind, it was just: "No, this is the best thing that can

[happen, that I die].'" (Interviewer: 'Why did you want that?') 'Because I was a burden to everyone.'–*Lisa*

Relatives were frustratingly powerless before the ravages of the disease. In the case of children with MS, the burden of having full insight into the most horrific possible outcome of the disease was primarily the parents'. The future of their children was shrouded in an agonizing uncertainty, with questions as to whether their child would be able to become physically independent, make a living of their own, or even survive. Sharing the difficulties could also have positive consequences, strengthening the ties within the biological family. The relationship with a partner was however often disturbed. As the disease came to dominate life more and more, the space for nurturing the relationship was reduced, and drifting apart was hard to avoid. Also, the negative impact of the disease on the participants' mood, as a result of despair and uncertainty, put the relationship to the test.

'We had a different attitude to each other. During this whole period, which was rather tough, the relationships change drastically, like not knowing if you'll be able to walk today. The uncertainty preys on you. And sure, I'm quite a happy person now, as I was then, too, but it takes a toll on your mood. You're not exactly happy as can be when you get this darn disease.'–*Amy*

Professional work and schooling were severely disturbed, due to both physical and cognitive disabilities, as a consequence of the frequent relapses. Some were dismissed from their jobs. The relapses became a constant threat to the demands of everyday life.

'So I missed out on a lot in school, since I was so tired. And then I had relapses. I didn't know from day to day what I would feel like, so I'd wake up in the morning and go: "Can I go or not?"'–*Tiffany*

As the disease progressed with no sign of abating, despite the best available treatment at the time, a consuming uncertainty grew. Each new day meant a risk of losing their abilities, the participants losing a bit of themselves, and ultimately losing their life. The apprehension of getting new symptoms was a greater concern than stationary impairments which to varying extents could be compensated for, adapted to, and eventually accepted. Once again, it was the uncertainty that caused most anxiety.

'I usually don't complain; I usually accept a situation for what it is. But when you don't know from one day to another if you're going to get out of bed, or if you're going to see anything, or hear anything. . .'–*Daniel*

When the next relapse came, as a setback, the participants nearly lost hope and despaired.

'And then, in December 2005, I woke up one Saturday morning, the 3d of December: "Oh no, my legs are tingling again! Damn! I can't take it, not once more!"'–*Lisa*

## Theme 2: A new treatment–a possibility for a new life

AHSCT as a treatment option was not brought up until conventional treatment strategies were exhausted. Undergoing this procedure for MS was at that time still considered experimental, and the first AHSCT for MS carried out in Sweden attracted a lot of media attention. Several

of the participants had either heard about this treatment themselves, or from relatives and acquaintances. In the choice between AHSCT and continuing with the ongoing disease-modifying treatment, which clearly was not enough to halt the aggressive disease course, the former appeared to be the only option.

'It was kind of like my last hope. For I had tried all available MS drugs. There were injections and all sorts. And then [my physician] told me: "Well, there is this alternative, but it is not one hundred percent." I remember saying: "No, but just do it!" Because I felt that I had no choice. [. . .] I was only about 21 when I got my diagnosis, and what would I be [at the end of the treatment]? 26, I think. Well, it felt like: "I've got nothing to lose."'–*Nicole*

Information about the potential adverse events of AHSCT, especially the risk of a lethal outcome, was emphasized by the healthcare providers. But even if there was a risk of dying from AHSCT, it was a risk worth taking.

'I just wanted to do it [AHSCT], I think. Since I felt so bad. For there was a risk that one would die, but. . . It wasn't a problem [laughs]. Or, well, I did not care about the risk. [Short pause]'–*Tiffany*

The thought of risking death as a consequence of the treatment appeared less frightening than letting the disease progress in the current way. Given the poor prognosis, life as it appeared at the time did not seem worth living–hence the inclination to take a significant risk. There was nothing to lose.

'So I felt, I wasn't scared or anything when he started talking about, like, there were, well, dangers, but I, like: . . . "I'll do it, since I've got nothing to lose." Since it was just getting worse and worse.'–*Melissa*

Family and friends were highly supportive in the participants' decision to proceed with the treatment, although it was emphasized that the final decision was for the patient alone. Supportive and thoughtful comments from health care personnel could also be decisive.

'Basically, I just heard the doctor say, "Well, you may die from this." So I was to die *again*! [First from MS, and then from the treatment.] "What the hell!" So I went . . . or they took me into my [hospital] room, and I just sat there crying. A nurse came and [asked me]: "But why are you crying?" I said: "Now I will die again." And she said: "Is that what the doctor told you?" I said: "No. . . but *basically*. . ." [The nurse]: "But what did he *really* say?" And I said: "No, but it [AHSCT] is *dangerous*!" And she said, "Mm . . . but how good is your life *now*?" And I: "Not very good" [. . .] So I [thought], like: "Well, this life is hardly an option." So I rang the bell. "Yes, call [the doctor] and say that I *want* to do it."'–*Lisa*

For lack of other treatment options, AHSCT offered hope, despite the sparse clinical experience. Mainly, the participants hoped for a stabilization of the disease course and a halt of the frequent relapses that lead to progressive disability. There was also hope for an improvement of the bodily dysfunctions, although the ultimate goal was to avoid a seemingly inevitable death due to MS. Consequently, AHSCT gave hope of a second chance, the chance of a new life.

'I was on my way out [of the hospital] when my parents were to take me for a walk [in the wheelchair] as usual [laughter], the same damn turn every day, and [the physician] came up

to us, took hold of my legs, and said: "My hope is that you'll be able to *stand* on your own legs one day." That is where we were, in our minds. "Imagine to be able to *stand up*, one day!" Anything else would be [a bonus]. First, that I would get to *live*, but then all this [almost all symptoms being gone], this was: "Wow, that is quite some bonus!" That I can [do] *everything* today. [. . .] The important thing then was to *halt* the disease.'–*Jennifer*

### Theme 3: Stem cell transplantation–a transition

After the decision of AHSCT was made, a number of prescribed measures was necessary before the treatment itself could start. Those included fertility preservation with cryopreservation of oocytes for the women, dental clearance, gastrointestinal examination, and the placement of a central venous access catheter. Waiting, impatiently, for the treatment to start could be agonizing for the participant.

'I did not find the treatment very tough, but waiting for it [to begin] was tough.'–*Jennifer*

Although AHSCT was not a routine treatment for MS, it was a common procedure for many other diseases. By accepting to undergo the treatment, the participant temporarily handed over the responsibility for their life to the health care: a submission based on trust. The patient's confidence in the health care system in general, their conviction that they would receive the best available professional care, and their personal trust in the treating physician, were profound.

'When I arrived at the hospital [the staff said]: "This is just another Monday to us. To us, it is not out of the ordinary." This was reassuring to hear. It made me [calm down].' (Interviewer: 'So you felt rather safe in that. . . during the treatment, then?') 'Yes, exactly, absolutely. Yes, definitely. And also with [my treating physician], who is a wonderful physician, too. I came to trust him, very much. And that made us feel safe, for me.'–*Amy*

When ten years had passed, memories from the AHSCT procedure itself were often diffuse and fragmentary, especially for the participants who were children at the time. Some episodes and events have been retold to the participants by family and relatives; other memories were rediscovered, for example by reviewing diary notes.

'Since I don't remember very much, a lot has been told to me instead: what happened, in what order. For to me, it is a bit of a mess. [I] have images, stills, rather than whole memories.'–*Brandon*

The first part of the treatment was mainly described in more positive terms; sharing experiences with other patients was especially valuable. The acute and transient adverse events of AHSCT were most obvious in the latter part of the procedure, as expected, and included abdominal pain, eating difficulties, nausea, vomiting, pain, panic attacks, sepsis and fatigue. Some were vividly described, with memories of dramatic scenes, others hardly remembered. During the period of hospitalization, a feeling of surrender and transfer of responsibility to the health care dominated, rather than fear. After the procedure, however, the emotions could catch up with the participants when they recollected the events.

'Well, when I was *in the midst* of it, I wasn't scared. Then, it was just like [whispering]: "Yeah, but do what [the hell you want]." Then comes [the fear], like when you wake up, and

go: "Ooh! Ooh!" [Like shivers.] Sort of. As when I read my diary only a week ago. There it says that I even had sepsis, and a temperature of 42 degrees [Celsius] when I stayed here. But I don't sort of remember this.'–*Lisa*

The contrast between being in perfect health before the start of MS, and being severely disabled, both by the disease and by the treatment, was frightening and confusing. Having been in total control of themselves, with full freedom of action, the participants became vulnerable, restricted, and exposed, lacking control.

'And at that time [in hospital during AHSCT], it hit me how MS. . . That I, who had been in full health, it really hit me that. . . when I was attached to tubes and everything, and was in pain: that I could not go [anywhere], that I was really *so sick*.'–*Christopher*

As a bridge between the precipitating state of illness and the following period of recovery, the period of confinement to the hospital during the AHSCT was described as a passage; something that one simply must get through in order to move forward.

'But all I remember, from those five days or whatever, with cytotoxins, was mainly a kind of passage, yes, to see the toxins flow into my body. And then I slept, and I wanted the room to be dark. Well, I don't remember all that much.–*Jennifer*

Because the treatment drained the patients' strength, life had to be put on hold temporarily. As far as medically and logistically possible, the time for starting the procedure was adjusted to minimize the impact on schooling, for example, or to accommodate requests for certain holiday celebrations. This extraordinary pause in life came to symbolize a transformation, a turning point, heralding a new and brighter chapter in life.

'And it felt almost like one fell asleep sick, and everything was on hold, and one woke up–I would not say well, because I did not know that then–but I did know that there was a great chance that it could turn out damn well. . .'–*Daniel*

### Theme 4: Reclaiming life

Both the state of illness and the procedure with AHSCT instilled a sense that everything was at stake, including life itself. Thoughts of death were recurring, both that the disease itself, but also that the treatment could lead to death. In light of the subsequent period of recovery, the treatment appeared as a turning point, a journey back to a normalized life.

'This is how it was: "if this goes on, well, who wants a life like this? And if it turns out badly [and I'll die from the treatment], well, *then it does*." So: "let it happen. Because something has to change." [Pause] [. . .] And then I thought, too, that "at least it's good if it happens down there [that I die in hospital], so that they don't have to find me [dead] in the morning, at home in bed [from MS]." [Laughs] *Strange* thoughts like these. [Pause] And then [after the treatment] I began the journey back.'–*Lisa*

The time after AHSCT was characterized by recovery and rehabilitation. For many, previously lost functions returned, to a varying extent and at varying speed. Some neurological improvement could also occur, already after the initial mobilization, when a lower dose of

chemotherapy was given. Especially for those who had very limited mobility at the time of the treatment, every little improvement made a significant difference.

'So then I had "the progress of the day", basically. Every day, there was progress [. . .] Simple things happened, like being able to change my position in bed, and, eventually, starting to walk again.'–*Brandon*

Aids, such as crutches and wheelchairs, which had previously been needed for ambulation, could gradually be abandoned.

'I was in a wheelchair at the beginning [after treatment]. So it was like. . . just to be able to get up from it, not needing it anymore. . . –*Melissa*

The improvement was occasionally described in summary, describing how the recovery of the neurological impairments resulting from MS very quickly regressed. An intuitive feeling of having been cured arose.

'And then I became really healthy. Or. . . it sort of disappeared. Such a great relief! I was not dizzy anymore–it disappeared! And no fatigue, and. . . It turned out really great.'–*Tiffany*

Even symptoms that had existed for a longer period before the treatment could regress, which nobody had dared to hope for.

'So in this way it [the treatment] exceeded [the expectations], since I had in fact had the symptoms for a long, long time, and it actually receded.'–*Daniel*

As neurological functions improved, previously lost and sorely missed skills and abilities could be regained. From being restricted to ambulate with the assistance of a wheelchair, walking frame, or crotches, the ability to walk by themselves, or even to run, was not only a milestone in the rehabilitation but also a strongly emotional and powerful recovery of independence and autonomy.

'I loved sports and had missed doing it. So the first thing I did was to start, like, doing sports again, and I remember [laughter], the first time I went running in the floodlit jogging track. . . [starts crying]. Look, I'm crying because it is such a nice memory! Well, but I *ran*! So I had music [in earphones], and I ran to Bruce Springsteen's "Born to Run." And it felt beautiful, kind of. . . [laughing] [. . .]. It felt cinematic, in a way. Like: "Here I come!" It was like a comeback.'–*Sarah*

Step by step, different domains of life were reclaimed and recaptured. Joy was a strong motivating force for the participants, but also a will to prove to themselves and others that they were not condemned. For the children, the period of severe illness, and to some extent also the treatment, meant that schooling was affected. However, with special support from teachers, they were able to catch up. The interruption could also provide an opportunity for a fresh start, and create the opportunity to choose a new direction in studies or professional life. Often, the way back to working life was by a gradual return, for example through initial period of so-called 'work training', a type of occupational rehabilitation, which gave a temporary relief from ordinary productivity requirements. Recapturing previously lost domains in life meant that that the normality previously lost was gradually regained. With this came a sense that not only external life was normalized, but also the participant's very identity, their self.

'So that fall was spent healing, kind of like becoming [*Jennifer*] again, I think. And then, when I returned to work in January, I think, or *they* got the impression that this is *me* returning. And then, to be sure, I returned [to normal work hours] at breakneck speed, also at work. Well, perhaps a bit too quickly [laughter] because it was so pleasant to return to. . . to *life*.'–*Jennifer*

The rapid and dramatic evolution of events leading to treatment with AHSCT made it hard for the participant to grasp the situation and to keep up emotionally. A great joy following treatment and the resolution of symptoms could escalate into euphoria which, in retrospect, was not felt to be in harmony with the patient's sense of self. Periods of sadness and lack of energy did also occur.

'I believe that when I was discharged from here [the hospital], after the treatment, if we skip a little bit in my past, I think my mind was probably in the same phase as when I had just become ill. So I was not really keeping up during the first year.'–*Daniel*

There could also be disappointment with the treatment outcome, in terms of the rate and extent of symptom regression, especially in comparison to other patients with a more rapid or more complete recovery. Criticism could be discerned regarding the timing of the treatment; could an earlier initiation of AHSCT have resulted in a more favourable outcome? Even though the outcome of the treatment did not always lead to the improvement that was expected, none of the participants expressed regret about having undergone AHSCT.

'But then it did not happen. . . I did not recover as quickly as most of the others probably did. I am probably the one with [most symptoms remaining after treatment] since I still use a wheelchair. But I got, I would, I got the stem cell transplantation a little late, which. . . You should not wait too long, so that the scars become permanent.'–*Christopher*

Orientation in a new mental landscape after completed treatment was not just demanding for the participants, but also for close relatives. As the participants' neurological functions improved and conditions changed, it could be a hard balancing act for relatives to support the recovery on one hand, and give space for the participants' increased autonomy, on the other. Relationships were put to the test.

'I had a boyfriend at the time, during my [treatment]. When I started to recover, I was still sick to him. And I could not cope with this, because he, sort of: "Now you need to rest, now you have to. . ." and so on. And I said: "No! I'm getting up now!"'–*Lisa*

## Theme 5: A bright future accompanied by insecurity

Although a considerable improvement, even a total regression, of neurological symptoms could follow the treatment with AHSCT, other symptoms would remain to varying extents, including concentration and memory difficulties, bladder dysfunction, fatigue, nausea, and impaired gait.

'I still struggle with concentration, and a little bit with memory [difficulties] and these things. But I have still found ways to cope.'–*Lisa*

When the remaining symptoms did not get worse, and no new ones arose, it became easier for the participants to accept and find strategies to overcome impaired functions. This stability

was highly appreciated and constituted a contrast to the preceding unpredictable disease course. Despite remaining impairments–some of them even relatively disabling, such as impaired gait–the participants still perceived that they were recovered, in full health. The disease was no longer a threat.

'I feel that I have recovered and am in full health, but in a sense, I am disabled, like walking. It can be hard if you have to do your shopping in half an hour, or something like that. (Interviewer: 'When you say in full health . . . Can you tell me more?') Yes, I feel incredibly strong and healthy in other respects. I can think clearly. So I feel that I'm well, which makes it a bit frustrating that my legs don't . . . [laughter] that I can't walk.'–*Christopher*

The sudden halt of the aggressive disease progression after treatment with AHSCT evoked a sense of having escaped an otherwise predestined, severe suffering, maybe even having escaped death. When considering the consequences if AHSCT had not been offered, had they lived in a region where this treatment option was unfamiliar, for example, the participants felt both fear and gratitude. Being able to share their experiences was important.

'I think I may have died if I had not been given this treatment. So just the thought that . . . and then, the fact that it turned around so quickly [laughter], and one had to keep up emotionally! So to get to . . . both work out and then to talk about it all, was really very important.'–*Jennifer*

The feeling of gratitude could result in a desire to seize every opportunity in life, as the treatment had led to a 'second chance', for example, to live an extraordinary life, to travel to foreign places, to complete a higher education, and to have a successful career. Relatives and acquaintances may have had such expectations on behalf of the participants, but it was primarily the participants' own desire. Failure to live up to these expectations was regarded as shameful.

'You want to feel like this: "Oh, I recovered, I'm well! Now I want to do everything!" [. . .] But this is what I think affected me quite a lot: "You were given a new chance," from being really ill, to being well, and it all happened so quickly. [. . .] And all the time I've felt, kind of: "Oh, I'll make up for what I lost out on in life," but I don't need to do that at all! I can just live.'–*Sarah*

When the participants' focus shifted, gradually, from the disease to everyday life, with relationships, family and professional work, they developed a peace of mind.

'After this treatment, my life is back to normal again, and my mind and memory are filled with things just like other people's. [. . .] It's about everyday life. . . what you'll do three weekends from now, what you are going to buy someone for their birthday, and so on.'–*Daniel*

Immediately after the treatment, the thought of re-emerging disease activity was a source of considerable concern. The feeling of an uncertain and unpredictable existence lingered. The clinical follow-up visits and magnetic resonance imaging were events when this uncertainty could become more concrete, but simultaneously providing a sense of security. The fear of new relapses could result in an increased vigilance, leading to a mis- or over-interpretation of unsignificant bodily symptoms.

'But at that time, at the beginning, I felt every little thing, sort of. "Oh, what if it is a touch of something! What if it is something! Ooh!"'–*Melissa*

As time went by, the fear of re-emerging disease activity decreased, and time itself was seen as a deciding factor for diminished concerns.

'So I'm totally convinced that there will be no change. Once this much time has passed, too. Like I said, it's been ten years. And during this ten-year period, nothing has happened.'–*Amy*

The need to tell others about their experiences of the disease, and of the treatment that the participants had at first, faded away over time. When meeting new people, the participants could choose whether nor not to tell them about the diagnosis. It was not considered necessary to mention the diagnosis at a job interview, for example. Others preferred to talk more freely about their diagnosis, in order to create an understanding and acceptance of hidden disabilities. Not infrequently, the participants could be met by surprise and supportive curiosity when they gave an account of the treatment and the good outcome.

'So I feel, kind of. . . No, I'd rather be open and tell what it is like, really. So I prefer that people know, that is, that I have a disease, rather than. . .'–*Nicole*

For three of the ten participants, the fear of a return of active disease activity came true; two experienced relapses, and another one had a new lesion on magnetic resonance imaging, without symptoms. Immunomodulating treatment was restarted. Not expecting a cure against the disease, the participants had created a certain mental preparedness for this situation, which made it easier for them to cope. The fact that the relapses were mild and transient, and that restart of immunomodulating treatment was enough to stabilize their disease at this stage, mitigated the setback.

(Interviewer: 'How did you feel when this examination showed that it was a relapse, the first one after the treatment?') '[Pause] It is really harder to answer than you'd think, because. . . I still did not have this, like, expectation [to be cured by AHSCT], but I did hope. And of course, I did feel *something*, but I can't quite put my finger on it. Of course, it was a bit of a dip, that "Well, even MRI [magnetic resonance imaging] showed a change, a new [lesion]"; it wasn't just a feeling that could not be confirmed.'–*Daniel*

The participants' attitude to the MS diagnosis after treatment differed. Some of the participants, without disease activity after AHSCT, expressed clearly that they did not have the disease anymore, or that they considered themselves to be in full health.

(Interviewer: 'I noticed that you say, consistently, "when I became healthy"'.) 'Mm.' (Interviewer: 'Do you want to elaborate a little?') Sarah: 'I became healthy! [laughs] Yes!' (Interviewer: '[laughs] Yes. . . So if someone were to ask you if you have MS, you would. . .?') 'No, I see myself as fully recovered. And this happened, boom, overnight, from ill to healthy.'–*Sarah*

Others expressed themselves more cautiously, in terms of having 'dormant MS', or that they 'did not have an active disease' anymore. Even though the disease was no longer a threat, it was still present. Remaining disabilities could be a reminder of the disease, as could follow-

up visits and magnetic resonance imaging. All had a conscious approach to the diagnosis of multiple sclerosis and kept reflecting on their attitude to the diagnosis, often with inner monologues and ambivalence.

'I used to say that I had *dormant MS*. But nobody believes it. "You *do*!?" "Yes." And this is how I want it to be, that you should not be able to tell. But I am constantly aware of it–I call her Märta-Stina [an old-fashioned Swedish name]–because it sounds like a rather harmless mask. Yes, she is my companion, since I cannot deny that she has existed, of course.'–*Lisa*

Despite great hardships, the unique experience of having multiple sclerosis and having been treated with AHSCT was emphasized in the interviews. Having contact with others who shared the same experiences was described as particularly valuable.

'So I've got vast experiences and learned to cherish life more, or something like that. I'm thinking, it is not always to be taken for granted. . .'–*Tiffany*

In spite of many years without signs of an active disease, there could still be a lingering sense of insecurity whether the disease would return in the future, especially as there are no extensive, long-term outcome data of AHSCT of multiple sclerosis. There are also no formal criteria for classifying a definite cure.

'I've had [the fear of MS symptoms returning] at the back of my mind, a little bit. This is one of the things that started waning lately, but still there is, sort of, you . . . Since one cannot label oneself as "cured", as I understand it, there is always like a concern that it may come back.'–*Brandon*

Even participants who considered themselves to be recovered did not refer to themselves as having been 'cured' from multiple sclerosis. There was a demand for such a classification, but the initiative was handed over to the health care profession.

'If [my treating physician], for example, would say . . . keeping in mind that he is very hopeful, but still, if he were to say: "But now [slams their hand in the table] . . . now I consider you cured!" Then I would probably buy it.'–*Jennifer*

## Discussion

In this qualitative interview study, we have reported lived experience of people with aggressive MS treated with AHSCT. At the onset of disease, it caused alienation and loss of the sense of control. All aspects of the participants' lives changed their lives were dominated by a deep uncertainty. In the absence of other treatment options, AHSCT offered hope. The treatment itself was described as a transformative passage where life had to be temporarily suspended. After AHSCT, previously lost functions returned, with recovery of autonomy, identity, and a sense of normality in most of the participants. As time went by, the participants' fear of re-emerging disease activity decreased, and the previously dominating uncertainty waned. All participants had a conscious approach to the diagnosis of multiple sclerosis; some expressed that they did not have the disease anymore. There was a demand for a classification of cure.

RRMS can cause a wide variety of neurological symptoms, and the initial manifestations of the disease can easily be overlooked or misunderstood, as reported by the participants in the present study. A strange feeling of bodily abnormality was sensed, such as experiencing a

peculiar numbness. Also, inexplicable difficulties were encountered when performing ordinary activities, such as playing tennis; the body did not respond as expected. Occurring before the actual diagnosis, these experiences could be described as 'bodily uncertainty' and a sense of being a 'different body', using terms by van der Meide *et al.* [24], representing the first experiences of an alienation due to MS.

Historically, the natural history of MS has not provided newly-diagnosed persons with MS much consolation or encouragement; after a period of relapses, a progressive phase follows eventually, with an unalterably advancing loss of physical and cognitive abilities [25]. About one of ten develop an aggressive disease course, associated with a rapid development of severe and irreversible disabilities [3].

With such a pre-understanding of MS, it is understandable that the participants in our study described their diagnosis as a traumatic and life-changing event. Receiving an MS diagnosis is a well-studied topic in the qualitative MS research field, consistently described as a trauma with subsequent shock, denial, anger, and fear, often due to an excessively negative understanding and poor general knowledge of the disease [26]. Relating to the diagnosis as a 'doom' could imply an anticipation of unnegotiable and uncontrollable agony, resulting in a gradually circumscribed freedom and a restricted existence. Fears of losing autonomy and independence due to MS is previously reported [27], which corresponds well with the view on MS diagnosis as a 'doom' or a 'sentence'.

Nearly all aspects of life were affected by the disease, including professional work, schooling, spontaneous everyday tasks, and relationships to family and friends; if not directly affected by current physical or cognitive impairments, there was an indirect impact due to fear of future disabilities.

As the disease advanced and abilities such as gait, sight, hearing and speech were impaired, the participants tried their best to adapt to the constantly changing conditions. Contrary to what one might assume, the existing disabilities, which could be compensated for and adapted to, were not their main concern, but rather the unknown disabilities to come. Under the constant threat of relapses and disease progression, the participants experienced a profound insecurity and uncertainty in most domains of life; everything could swiftly be lost, even life itself. Uncertainty has been widely recognised as a grave concern of MS-patients, contributing to mental distress [28–30] and dominating the understanding of 'prognostic risk' in people newly diagnosed with MS [31]. From uncertainty, a lack of control follows: lack of control over one's body and everyday life, but also of the risk of future economic insecurity and social isolation [32, 33]. Also, partners of people with MS suffer from an unpredictable future and uncertainty as a result of the increased disability of their loved ones, and subsequent restrictions of their own personal freedom and self-actualization [34]. Three participants in our study were children at time of the diagnosis. In their case, it became particularly evident that MS is not only a disease affecting the person with the diagnosis, but the whole family, and especially their parents. The whole existence of the family pivoted around the effects of the dramatic disease course, and the participants could sometimes sense the agonizing uncertainty of their parents: would their child survive to adulthood, and if so, would they be able to live an independent life and make a living of their own? Parents' multifaceted uncertainty in relation to the illness and to the future of their child, and their coping strategies, are described in previous studies, which correspond to the findings in our study [35]. The extraordinary state with a child being ill can also have a positive side-effect in bringing the members of the family closer together, as reported in the present study. A recent review on patients' and parents' perspectives on paediatric MS also concluded that the patient's families have the potential not only to adjust to the situation, but also to be strengthened by common coping strategies and supportive engagement [36].

Generally, people with MS have often developed a strong illness identity, with a legitimate sense of not being able to control the disease and its consequences [37]. As disabilities gradually increase, and social interactions consequently change, the illness progressively merges with identity [38]. With respect to the unusually aggressive disease progression in the present cohort, the participants expressed having a lack of time and disease respite to adequately process the recurrent crises and adjust life accordingly. Instead, an increasing sense of alienation developed. The identity change that follows diagnosis and disease manifestations of MS have previously been thoroughly described in terms of altered 'sense of self' [26], 'self-identity' [38], 'perceptions of self' [39] or 'self-concept' [40]. Also, people living with a person with MS experience a change of identity [41]. Understanding the full range of the disease consequences for the individual ought to be valuable knowledge for healthcare workers when providing persons with MS professional care and guidance.

When AHSCT was proposed for the participants in this study, it was seen as the final option at a time when other available treatment options had been exhausted. The choice was then between undergoing an (for MS indication) experimental and possibly dangerous procedure for a chronic disease, or continuing the ongoing disease-modifying treatment, with an imminent risk of further neurological deterioration and even death. AHSCT offered hope where no other beneficial option was available. The participants in this study, or the parents of the three participants who were children at the time, considered it a risk worth taking. Many persons with MS seem to be willing to take a surprisingly high risk when undergoing treatment for MS. In a questionnaire study by Chacińska *et al.* in a Polish MS cohort, 81% would accept a treatment-related mortality of >1% if undergoing a potentially curative treatment; people with an aggressive disease course would tolerate an even higher risk [42]. The participants in our study were, to our knowledge, never promised a potentially 'curative' treatment. Rather, AHSCT was perceived as the only remaining treatment option. However, in accordance with the findings of Chacińska *et al.*, the participants in our study were willing to take a significant risk, though not promised a cure; treatment-related mortality following AHSCT for MS, according to a retrospective study from 2002, was reported to be 6%, which was the number presented to the participants in this study at the time of the decision for AHSCT, 2004–2006 [43].

In our study, experiences from the AHSCT procedure itself were more vaguely described in terms of a transition, and reported acute side-effects generally corresponded to toxicity previously described [14]. The period following AHSCT was described as a 'journey back' from a state of illness and alienation, characterized by disability and uncertainty, to a state of health, stability and healing. As previously lost domains in life could be recaptured and reclaimed, such as professional work and schooling, a sense of normality gradually re-emerged. Not only were the bodily functions restored, to various extents, but the very identity of the participants was recovered. The participant Jennifer captured and summarized this strikingly: 'I became Jennifer again.'

A major component in the sense of re-established health and normality was the sense of a gradually waning uncertainty, as the fear of returned disease activity became less prominent. With ten years passed, the participants related to the diagnosis of MS in various ways. Participants without any signs of disease activity could describe the disease as being 'dormant', 'not active', or themselves as 'healthy'/'fully recovered'/'in full health' (Swedish: 'frisk'). Some did not regard themselves as having MS anymore. However, no one described themselves as having been 'cured' (Swedish: 'botad').

The Swedish word 'frisk' that is used frequently by the participants is not easily translated into English. It is an antonym to 'sjuk', English 'ill', but also corresponds to the process of

healing, both 'recover from a disease' and 'be cured from a disease,' or, in other words, the transition from a state of illness to a state of health [44].

The strife for a 'cure' for MS has been sparsely discussed in the scientific literature, possibly even avoided, with some prominent exceptions. For example, in 2009, Weiner proposed three different definitions of a cure for MS: halting progression of disease, reversing neurological deficits, and preventing MS [45]. In 2013, in light of the outcomes after the introduction of the drug alemtuzumab for the treatment of MS, the Editorial of *Multiple Sclerosis and Related Disorders* proposed a working definition of cure as NEDA-3 ('no evidence of disease activity', defined as no new relapses, no new lesions on magnetic resonance imaging, and no disability increase [46]) sustained for 15 years without continuous disease-modifying treatment, thus in agreement with the first definition of cure proposed by Weiner [47]. The result of the present study can add two aspects to this discussion. First, it gives support to an understanding of cure (described in terms of 'being healthy', 'being in full health', 'not having MS anymore,' etc.) as an absence of new symptoms, an absence of neurological deterioration, and ultimately a waning uncertainty. Secondly, there is an expectation from persons with MS for health care professionals to establish criteria for a definition of 'cure', or a similar comprehensible terminology to describe the state of a long-lasting absence of disease activity, where no return of disease activity is expected.

In accordance with the changed prognosis and understanding of HIV/AIDS with the introduction of the highly active antiretroviral treatment in the late 1990s [48], AHSCT was described in terms of a second chance and an opportunity for a new life by the participants in the present study. However, the resulting expectations from the patients themselves, and from their friends and family, to live an extraordinary life when 'cured' from MS could be difficult. Too much emphasis on the possibility of a 'cure' could be misleading, as it might cause disappointment and unjustified risk-taking in treatment choice. For the participants, the risk of a return of disease activity evoked a sense of insecurity, although this feeling became less prominent with time. In the cases where disease activity returned, though not as aggressively as before treatment with AHSCT, not having expected a 'cure' was helpful in coping, and the experience of good health could still be perceived. On the other hand, absence of disease activity was not synonymous with experienced good health; disabling symptoms such as fatigue, pain, and walking difficulties could still persist and have a major impact on life.

This study has several limitations, primarily a low number of participants. With their unusual aggressive disease course, they are also not representative of the average MS patient. All were treated by the same neurologist, at the same hospital, within a limited period of time, and well aware of each other; some of the participants are even friends. It is therefore likely that the perception of the treatment outcome has been partly influenced by experiences of the other participants. Altogether, this makes claims for a wider transferability of the results precarious. With ten years passed since the treatment, memories can easily be distorted, important events subdued, and others reinforced to fit into a comprehensible and coherent narrative, resulting in a recall bias. Some participants were also more articulate than others, resulting in a richer material from them. The unstructured arrangement of the interviews was an active choice to avoid influence of the interviewers' preconceived notions as much as possible, and to give maximal room for flexibility, but this mode may also reduce the reliability of the data. Although the interviewers used several methods to reduce the risk of a response bias, for instance interfering in the narration of the participants to a minimal extent, formulating open-ended questions and avoiding leading ones, there is always a risk that the participants told their stories in a way that they thought would please the interviewers and fit into a 'success story' of AHSCT. Furthermore, the last portion of one recording was lost due to a technical failure.

Despite the small cohort, however, the results of the initial disease experience and the following uncertainty and identity change are remarkably coherent with previous qualitative research and appear to reflect the experiences of many persons with MS. Nevertheless, many aspects make the cohort of this study unique: the unusually aggressive disease course, the failure of treatment with more conventional disease-modifying treatments, the decision to proceed with AHSCT, the reversion of neurological disabilities and, consequently, the waning uncertainty and restored sense of self. The cohort was diverse, including both men and women, persons of different ages, with different treatment outcomes, and inclusion bias was prevented by recruitment of all patients treated with AHSCT during a certain time period. The methodology was, we believe, well-suited for this exploratory study approach. To enhance trustworthiness in the analysis process, several strategies were used, including investigator triangulation with prolonged engagement of the data, thick description of the context and the cohort, representation of all participants in the interpretation of the data, rigour of the analysis, and the use of a methodology well-known to the researchers [19, 49]. Ultimately, it is for the readers themselves to decide upon the reliability of the results and conclusions.

Based on the outcome of this exploratory study, future studies can be designed with topics for semi-structured or structured interviews, facilitating the inclusion of a larger number of patients. Preferably, interviews should be conducted repeatedly, in a prospective manner, starting at the time of decision for AHSCT.

To the best of our belief, knowledge about the experience of patients undergoing a treatment is crucial both for health care professionals and for people with MS when considering a treatment option, and also for designing a suitable and amenable follow-up. In this regard, we believe the present study makes a vital contribution to the field, by providing a unique perspective of long-term lived experience after treatment with AHSCT for MS. AHSCT was perceived in terms of a second chance and an opportunity for a new life. The treatment became a transition from a state of illness to a state of health, enabling a previous profound uncertainty to wane, and normality be restored. In a broader perspective, the results give a first insight into the experience of not having MS anymore following a highly effective induction treatment, thereby opening for a paradigm shift in the view of MS as a chronic disease with no possible cure.

## Acknowledgments

We would like to thank Jan Fagius, MD, PhD, for his vital and compassionate contribution to this study and to our translator, Anna Uddén, PhD, for her highly qualified work with the text and for fruitful discussions.

## Author Contributions

**Conceptualization:** Andreas Tolf, Joachim Burman, Gullvi Flensner.

**Data curation:** Andreas Tolf, Gullvi Flensner.

**Formal analysis:** Andreas Tolf, Helena Gauffin, Anne-Marie Landtblom, Gullvi Flensner.

**Investigation:** Andreas Tolf, Helena Gauffin, Joachim Burman, Gullvi Flensner.

**Methodology:** Andreas Tolf, Helena Gauffin, Anne-Marie Landtblom, Gullvi Flensner.

**Supervision:** Helena Gauffin, Anne-Marie Landtblom, Gullvi Flensner.

**Validation:** Helena Gauffin, Joachim Burman, Anne-Marie Landtblom, Gullvi Flensner.

**Writing – original draft:** Andreas Tolf.

**Writing – review & editing:** Andreas Tolf, Helena Gauffin, Joachim Burman, Anne-Marie Landtblom, Gullvi Flensner.

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
