## [Decision Letter · Decision Letter 0]

23 Oct 2023

PONE-D-23-23564Experiences of being treated with autologous haematopoietic stem cell transplantation for aggressive multiple sclerosis: A qualitative interview studyPLOS ONE

Dear Dr. Tolf,

Thank you for submitting your manuscript to PLOS ONE. After careful consideration, we feel that it has merit but does not fully meet PLOS ONE’s publication criteria as it currently stands. Therefore, we invite you to submit a revised version of the manuscript that addresses the points raised during the review process.

The manuscript has been evaluated by two reviewers, and their comments are available below.

Both reviewers request additional information (see comments below). I appreciate that some of these requests may not be possible if you do not have the relevant data. One of the reviewers also requests qualitative reporting guidelines However, as you have already included completed copies of the COREQ and SPQR, I do not feel that any additional such tools are necessary. Could you please revise the manuscript to carefully address the concerns raised?

We look forward to receiving your revised manuscript.

Kind regards,

Steve Zimmerman, PhD

Senior Editor, PLOS ONE

Reviewers' comments:

Reviewer's Responses to Questions

**Comments to the Author**

1. Is the manuscript technically sound, and do the data support the conclusions?

Reviewer #1: Yes

Reviewer #2: No

2. Has the statistical analysis been performed appropriately and rigorously? 

Reviewer #1: N/A

Reviewer #2: No

3. Have the authors made all data underlying the findings in their manuscript fully available?

Reviewer #1: No

Reviewer #2: No

4. Is the manuscript presented in an intelligible fashion and written in standard English?

Reviewer #1: Yes

Reviewer #2: No

5. Review Comments to the Author

Reviewer #1: The Authors present the results of a qualitative interview study exploring the experience of 10 patients with MS who had been treated with AHSCT on average 10 years earlier. The study is original and provides valuable insight into patients’ perspectives on such demanding procedure. I believe this study is of interest to the MS specialist, as it explores several MS-related issues and discusses relevant topics, such as the delay that patients usually perceive in being referred for AHSCT, and the potential deleterious effects that such delay may have had on their outcomes.

I have minor suggestions only.

1. Please add EDSS at baseline and the 10-y follow-up in Table 1.

2. As few patients who did not observe any improvements are mentioned, I suggest expanding this section, shall there be enough material in the interviews. Among cases that continued to worsen despite the procedure, did someone manifest any regrets about the decision to undergo AHSCT?

3. Lines 579-580: I suggest rephrasing the sentence “After AHSCT, previously lost functions returned, with recovery of autonomy, identity, and a sense of normality” specifying that such improvement happened in most cases (but not in 100% of the patients treated).

Reviewer #2: In this study, Andreas Tolf and colleagues present the results of qualitative interviews with 10 participants who underwent AHSCT at a single center in Sweden. Themes were identified from these interviews using an inductive approach. It is an important study topic, since patient perspective and post-transplant well-being is crucial to explore for such an aggressive therapy. The selection of quotations from the participants and the overall flow of the article are well written, however I have some suggestions to consider for possible improvement.

The sample size of this study is small, it would be more compelling if other centers were approached to contribute qualitative information to ensure that the themes of responses are generalizable.

There are, of course, very different mobilization, conditioning, and graft-processing protocols with AHSCT, and the protocol used in this center was not described. The difference in intensities should be outlined. In addition, relating participant responses to pre-transplant disease activity, post-transplant disease activity, disability worsening, or side effects such as infection or length of hospitalization would be important to include. A quality of life or patient reported outcome questionnaire should be included as well to correlate with findings, there have been multiple studies of AHSCT that has shown a positive effect from Transplant.

The methodology of conducting the interview is detailed, however the analysis was not clear. Simply stating that “Consensus on the coding, the latent content, and the level of abstraction was reached through exhaustive virtual meetings with the three coding researchers” is insufficient. There are software programs such as NVIVO for qualitative data analysis, as well as criteria for evaluating qualitative studies and data that should be employed. This may allow for figures to be generated and help the article be visually appealing. At minimum, EQUATOR network reporting guidelines for qualitative studies should be included. The article should also be proofread and edited in formal English, there are different fonts used in this PDF which may be a small oversight but should be corrected.

6. PLOS authors have the option to publish the peer review history of their article (what does this mean?). If published, this will include your full peer review and any attached files.

Reviewer #1: **Yes: **Alice Mariottini

Reviewer #2: No

---

## [Author Response · Author response to Decision Letter 0]

6 Dec 2023

Answer: We have now formatted the headings and table titles according to the attached instructions; these changes have not been marked in the document "Revised Article with Changes Highlighted".

2. In your Data Availability statement, you have not specified where the minimal data set underlying the results described in your manuscript can be found. PLOS defines a study's minimal data set as the underlying data used to reach the conclusions drawn in the manuscript and any additional data required to replicate the reported study findings in their entirety. […] If there are ethical or legal restrictions to sharing your data publicly, please explain these restrictions in detail.

Answer: We have, as requested, updated the data access information with non-author contact information (Excerpts of the transcripts relevant to the study is available upon request from Head of Department, Department of Medical Sciences, Uppsala University, SE-751 85 Uppsala, Sweden). We hope that our Data Availability statement can be updated on our behalf to reflect this information, as suggested in the mail response to us November 15, 2023.

3. Please review your reference list to ensure that it is complete and correct. 

Answer: We have reviewed the reference list and found no inaccuracies or articles that have been retracted. No changes to the reference list have been made.

 

Reviewer 1

1. Please add EDSS at baseline and the 10-y follow-up in Table 1.

Answer: We have added EDSS at AHSCT and at the time of the interview in Table 1, as suggested.

2. As few patients who did not observe any improvements are mentioned, I suggest expanding this section, shall there be enough material in the interviews.

Answer: Thank you for this question, which shows that our medical description of the cohort was not sufficient. In fact, all participants improved after the treatment, even if not everyone improved as much as they had hoped for before the treatment. The improvement is reflected, among other things, by the decreasing EDSS values, as shown in the following publication including exhaustive medical data on the same cohort:

Tolf A, Fagius J, Carlson K, Akerfeldt T, Granberg T, Larsson EM, Burman, J. Sustained remission in multiple sclerosis after hematopoietic stem cell transplantation. Acta Neurol Scand. 2019;140(5):320-7.

To clarify this, in addition to the EDSS values in Table I, we have added more clinical information in Table I, as also requested by Reviewer II: EDSS before AHSCT, annual relapse rate prior to AHSCT, length of hospitalization, acute side effects related to the treatment, EDSS at the time of the interview, as well as disease activity and any deterioration after AHSCT.

3. Among cases that continued to worsen despite the procedure, did someone manifest any regrets about the decision to undergo AHSCT?

Answer: None of the participants expressed any regret about having undergone AHSCT. On the contrary, one of the participants, ‘Daniel’, who actually had relapses after the treatment, described how his life, despite this, had returned to normality: ‘After this treatment, my life is back to normal again’. We have clarified in the text that none of the participants regretted the treatment by adding the following section:

Even though the outcome of the treatment did not always lead to the improvement that was expected, none of the participants expressed regret about having undergone AHSCT.

4. Lines 579-580: I suggest rephrasing the sentence “After AHSCT, previously lost functions returned, with recovery of autonomy, identity, and a sense of normality” specifying that such improvement happened in most cases (but not in 100% of the patients treated).

Answer: We have revised the sentence as suggested:

After AHSCT, previously lost functions returned, with recovery of autonomy, identity, and a sense of normality in most of the participants.

 

Reviewer 2 

1. The sample size of this study is small, it would be more compelling if other centers were approached to contribute qualitative information to ensure that the themes of responses are generalizable.

Answer: We agree that a significant limitation of this study is its small number of participants. By including more participants, the generalisability of the conclusions could have been strengthened. 

However, this cohort is almost identical to the first ten patients who underwent this treatment in Sweden. By including all centres that performed AHSCT against relapsing-remitting MS in our country, only one additional patient could have been considered for participation in the study without changing the time interval for inclusion in the study (2004-2007), or by including patients who underwent AHSCT for relapsing-remitting MS in other countries, which would have presented other challenges, mainly for linguistic and logistical reasons.

With the experiences from this study, we look forward to conducting future studies using qualitative methodology on patients who have undergone AHSCT for relapsing-remitting MS and to include more participants.

2. There are, of course, very different mobilization, conditioning, and graft-processing protocols with AHSCT, and the protocol used in this center was not described. The difference in intensities should be outlined.

Answer: We have added information on the protocols used and the differences in intensity between different protocols in the Methods section:

AHSCT can be conducted according to various protocols of differing intensities. These are commonly classified as high-, intermediate-, or low-intensity protocols. All ten participants underwent mobilisation of the haematopoietic stem cells with a single dose of cyclophosphamide 2g/m2 and filgrastim 5-10 µg/kg/day over 6-7 days, followed by collection of the stem cells on day 10 or 11 after the start of mobilisation. The graft was then cryopreserved without further manipulation. Nine of the participants (all except the youngest, ‘Brandon’) then underwent conditioning with the intermediate intensity BEAM-ATG regimen consisting of carmustine 300 mg/m2, etoposide 800 mg/m2, cytarabine 800 mg/m2, melphalan 140 mg/m2, and antithymocyte globulin 10 mg/kg. "Brandon" was treated with a low-intensity conditioning regimen consisting of cyclophosphamide 200 mg/mg and antithymocyte globulin 6 mg/kg.

3. In addition, relating participant responses to pre-transplant disease activity, post-transplant disease activity, disability worsening, or side effects such as infection or length of hospitalization would be important to include.

Answer: We share the view that such information can enhance the comprehension of the participants' responses. Therefore, in Table I, we have included the suggested clinical details: EDSS before AHSCT, annual relapse rate prior to AHSCT, length of hospitalization, acute side effects related to the treatment, EDSS at the time of the interview, as well as disease activity and any deterioration after AHSCT. 

4. A quality of life or patient reported outcome questionnaire should be included as well to correlate with findings, there have been multiple studies of AHSCT that has shown a positive effect from Transplant.

Answer: We agree that such a questionnaire could have been included in the results and provided additional perspectives through quantitative data. Unfortunately, a formal assessment of quality of life was not made at the time of the interview.

5. The methodology of conducting the interview is detailed, however the analysis was not clear. Simply stating that “Consensus on the coding, the latent content, and the level of abstraction was reached through exhaustive virtual meetings with the three coding researchers” is insufficient.

Answer: We are grateful for this opinion and have revised the description of the method section to better explain the steps of the analysis:

Qualitative content analysis was used for the analysis, according to the method described and elaborated by Graneheim, Lundman and Lindgren [19, 22, 23]. The transcribed interviews constituted the units of analysis. After multiple readings to grasp the entirety of the content and to ‘obtain a sense of the whole’ [23], the individual meaning units in the text were identified, representing a ‘constellation of words or statements that relate to the same central meaning’ [19]. In order to improve readability for further analysis, the meaning units were then condensed, transitioning from the verbatim transcription to a clearer, more formal expression, while preserving the original inherent meaning. Subsequently, these statements were examined and interpreted to establish their deeper meaning. This process, and interpretation of the latent content, as outlined above and exemplified in Table 3, was interpreted by three of the researchers independently (GF, HG, AT), using the software Microsoft Word (version 2016).

Interpreting the latent meaning is one of the most delicate steps in qualitative content analysis and requires an understanding of the entire context in which an individual "meaning unit" is situated. Often, such interpretation is intuitive and uncomplicated, but sometimes different researchers make different assessments and interpretations. This can involve perceiving a meaning unit differently or having differing views of the context. To strengthen trustworthiness, the analysis was therefore conducted by the three researchers entirely independently, as described above. In joint and exhaustive digital meetings, the individual analyses of the three interpreting researchers were then compared. In-depth discussions were held about important passages and interpretations that might differ, with the aim of achieving consensus on the coding, latent content and level of abstraction. AT kept a record and noted the interpretations of the latent meaning and codes that had emerged from the consensus process. At times, the audio recordings of the interviews were also played to clarify emphases and vocal expressions that could affect the interpretations. Through the joint discussions, a deeper understanding of both the individual statements, the codes, and the whole was also achieved. An example of the process with the different analysis steps is presented in Table 3.

From the deliberations in those sessions, GF and HG formulated themes and subthemes. A theme was perceived as a ‘thread of meaning’ [23] connecting latent content across various meaning units and codes, reflecting the text's latent content. The themes can further be subdivided into subthemes. [19] AT performed the re-contextualisation [23] by grouping the codes with their respective meaning units into the formulated themes and subthemes, using the software MAXQDA 2020 (version 20.4.2). Based on this compilation, representative citations were chosen to illustrate the themes and sub-themes (AT). Citations were translated from Swedish to English by an authorized translator with experience of translating both fictional and scientific literature, and nuances in the translations were discussed in sessions with the translator. Consensus on the final themes, subthemes, and appropriate citations was reached in a meeting with all authors.

6. There are software programs such as NVIVO for qualitative data analysis, as well as criteria for evaluating qualitative studies and data that should be employed. This may allow for figures to be generated and help the article be visually appealing.

Answer: We agree with the observation that the article is notably dense in textual content. During the re-contextualisation after the analysis, we used the software MAXQDA 2020 to structure the quotations. However, this software did not offer a satisfactory or clarifying graphical representation of our material.

7. At minimum, EQUATOR network reporting guidelines for qualitative studies should be included. 

Answer: In the reporting of this study, we have used the two recommended EQUATOR network reporting guidelines, the COREQ checklist and the SRQR guidelines, as reported in the first paragraph in the Methods section.

8. The article should also be proofread and edited in formal English.

Answer: In this study, in accordance with the practice for qualitative studies, we have chosen to present the participants' statements verbatim, without formalizing the language, except that the quotes have been translated by a certified translator from Swedish to English. The aim of presenting the participants’ statements verbatim has been to enhance credibility and transparency. The text that is not quotes is written in formal English and has undergone proof-reading by a certified translator. We would still like to apologize for any linguistic or grammatical shortcomings, which we have endeavoured to identify and rectify.

9. There are different fonts used in this PDF which may be a small oversight but should be corrected.

Answer: The text has now been formatted to a consistent font.

---

## [Decision Letter · Decision Letter 1]

9 Jan 2024

Experiences of being treated with autologous haematopoietic stem cell transplantation for aggressive multiple sclerosis: A qualitative interview study

PONE-D-23-23564R1

Dear Dr. Tolf,

We’re pleased to inform you that your manuscript has been judged scientifically suitable for publication and will be formally accepted for publication once it meets all outstanding technical requirements.

Kind regards,

Emily Lund

Academic Editor

PLOS ONE

Additional Editor Comments (optional):

Reviewers' comments:

Reviewer's Responses to Questions

**Comments to the Author**

1. If the authors have adequately addressed your comments raised in a previous round of review and you feel that this manuscript is now acceptable for publication, you may indicate that here to bypass the “Comments to the Author” section, enter your conflict of interest statement in the “Confidential to Editor” section, and submit your "Accept" recommendation.

Reviewer #1: All comments have been addressed

2. Is the manuscript technically sound, and do the data support the conclusions?

Reviewer #1: Yes

3. Has the statistical analysis been performed appropriately and rigorously? 

Reviewer #1: N/A

4. Have the authors made all data underlying the findings in their manuscript fully available?

Reviewer #1: Yes

5. Is the manuscript presented in an intelligible fashion and written in standard English?

Reviewer #1: Yes

6. Review Comments to the Author

Reviewer #1: (No Response)

7. PLOS authors have the option to publish the peer review history of their article (what does this mean?). If published, this will include your full peer review and any attached files.

Reviewer #1: **Yes: **Alice Mariottini

---

## [Editor Report · Acceptance letter]

30 Jan 2024

PONE-D-23-23564R1 

PLOS ONE

Dear Dr. Tolf, 

I'm pleased to inform you that your manuscript has been deemed suitable for publication in PLOS ONE. Congratulations! Your manuscript is now being handed over to our production team.

Kind regards, 

on behalf of

Dr. Emily Lund 

Academic Editor

PLOS ONE